# Anti-Allergic Effect of 3,4-Dihydroxybenzaldehyde Isolated from *Polysiphonia morrowii* in IgE/BSA-Stimulated Mast Cells and a Passive Cutaneous Anaphylaxis Mouse Model

**DOI:** 10.3390/md20020133

**Published:** 2022-02-10

**Authors:** Eun-A Kim, Eui-Jeong Han, Junseong Kim, Ilekuttige Priyan Shanura Fernando, Jae-Young Oh, Kil-Nam Kim, Ginnae Ahn, Soo-Jin Heo

**Affiliations:** 1Jeju Marine Research Center, Korea Institute of Ocean Science & Technology (KIOST), Jeju 63349, Korea; euna0718@kiost.ac.kr (E.-A.K.); junseong@kiost.ac.kr (J.K.); 2Research Center for Healthcare and Biomedical Engineering, Chonnam National University, Yeosu 59626, Korea; iosu5772@naver.com; 3Department of Food Technology and Nutrition, Chonnam National University, Yeosu 59626, Korea; 4Department of Marine Bio-Food Sciences, Chonnam National University, Yeosu 59626, Korea; shanurabru@jnu.ac.kr; 5Food Safety and Processing Research Division, National Institute of Fisheries Science, Busan 46083, Korea; ojy0724@korea.kr; 6Chuncheon Center, Korea Basic Science Institute (KBSI), Chuncheon 24341, Korea; knkim@kbsi.re.kr; 7Department of Marine Biology, University of Science and Technology, Daejeon 34113, Korea

**Keywords:** *Polysiphonia morrowii*, 3,4-dihydroxybenzaldehyde, immunoglobulin E, bone marrow-derived cultured mast cells, passive cutaneous anaphylaxis

## Abstract

In this study, we investigated the anti-allergic effects of 3,4-dihydroxybenzaldehyde (DHB) isolated from the marine red alga, *Polysiphonia morrowii*, in mouse bone-marrow-derived cultured mast cells (BMCMCs) and passive cutaneous anaphylaxis (PCA) in anti-dinitrophenyl (DNP) immunoglobulin E (IgE)-sensitized mice. DHB inhibited IgE/bovine serum albumin (BSA)-induced BMCMCs degranulation by reducing the release of β-hexosaminidase without inducing cytotoxicity. Further, DHB dose-dependently decreased the IgE binding and high-affinity IgE receptor (FcεRI) expression and FcεRI-IgE binding on the surface of BMCMCs. Moreover, DHB suppressed the secretion and/or the expression of the allergic cytokines, interleukin (IL)-4, IL-5, IL-6, IL-13, and tumor necrosis factor (TNF)-α, and the chemokine, thymus activation-regulated chemokine (TARC), by regulating the phosphorylation of IκBα and the translocation of cytoplasmic NF-κB into the nucleus. Furthermore, DHB attenuated the passive cutaneous anaphylactic (PCA) reaction reducing the exuded Evans blue amount in the mouse ear stimulated by IgE/BSA. These results suggest that DHB is a potential therapeutic candidate for the prevention and treatment of type I allergic disorders.

## 1. Introduction

The rapidly increasing incidence of allergic diseases, including asthma, allergic rhinitis, and atopic eczema, has become a significant public health problem [1,2]. Among allergic diseases, type I allergic reactions are the most common in allergic patients and are caused by immunoglobulin E (IgE)/antigen-mediated mast cell activation [3]. Therefore, the inhibition of mast cell activation is a crucial factor in the development of treatment strategies for type I allergic diseases [3]. Mast cells may be activated through the crosslinking of high-affinity receptors to IgE (FcεRI) by the IgE/antigen complex expressed on the surface of cells [4,5,6]. The activated mast cells are sensitized to release allergic mediators such as granular contents (β-hexosaminidase) as well as allergic cytokines (interleukin (IL)-4, IL-5, IL-6, IL-13), and tumor necrosis factor (TNF)(-α) and the chemokine, thymus activation-regulated chemokine (TARC), which induces allergic disease [6,7]. In particular, the major effectors of allergic symptoms are the allergic cytokines by regulated pathological problems on mast cells, as overexpression may cause pathological problems [5]. Recent studies reported that NF-κB signaling plays a key role in the regulation of the expression and production levels of immune, inflammatory, and allergic cytokines and mediators [7,8,9,10]. This suggests that the NF-κB signaling pathway is crucial in the allergic response. Various medications such as antihistamines or corticosteroids have been commonly used to inhibit allergic reactions; however, their use is limited owing to their severe or partial adverse effects such as damage to the immune system, cardiac toxicity, and bone loss [6,7,11]. In view of this, many studies have focused on the use of safe and natural materials as alternatives with lower toxicity and fewer side effects [9,10]. Among these natural materials, algae have various biological activities. Algae contain a variety of functional compounds with an intriguing structural diversity [12,13,14,15]. *Polysiphonia morrowii Harvey* (*P. morrowii*), a red alga, produces several bioactive substances such as 5-bromo-3,4-dihydroxybenzaldehyde (BDB), 3,4-dihydroxybenzaldehyde (DHB), 3-bromo-5-(ethoxymethyl)-1,2-benzenediol, 3-bromo-4,5-dihydroxybenzyl methyl ether, 3-bromo-4,5-dihydroxybenzaldehyde, and bis (3-bromo-4,5-dihydroxybenzyl) ether [12,13,14,15,16,17,18,19]. These bio-active substances have diverse physiological effects including anti-inflammatory, antiviral, anti-bacterial, antioxidant, antiobesity, and antiallergic activities [12,13,14,15,16,17,18,19]. In our previous study, we reported that BDB isolated from *P. morrwii* possess antiallergic effects in IgE-induced mast cell activation and passive cutaneous anaphylactic (PCA) reaction in mice model [12]. Additionally, recent studies reported that the DHB, a compound without a bromo group from BDB, has the antioxidant, anti-inflammation and antiallergic effects [17,19,20]. However, there are no reports elucidating the plausible biological mechanism related to the antiallergic effects of DHB in IgE-induced mast cell activation and PCA mice model. Therefore, in this study, we evaluated the antiallergic effects of DHB by inhibiting allergic factors and attempted to unveil the underlying biological mechanism in IgE/BSA-stimulated a bone marrow cultured mast cells (BMCMCs) and a PCA mouse model.

## 2. Results

### 2.1. DHB Effectively Decreased β-Hexosaminidase Release without Cytotoxicity in BMCMCs

To identify the cytotoxicity of DHB on BMCMCs, the MTT assay was performed. At all of the studied concentrations, DHB did not show any cytotoxicity in the BMCMCs compared to the untreated control cells (Figure 1A). On the basis of this result, we determined the concentration range of DHB for further experiments. Next, we investigated the effect of DHB on the degranulation of BMCMCs by evaluating the release of β-hexosaminidase in the IgE/BSA-stimulated BMCMCs. β-hexosaminidase release is a widely investigated biomarker of mast cell degranulation [21]. As shown in Figure 1B, DHB significantly reduced the release of β-hexosaminidase in the IgE/BSA-stimulated BMCMCs. These results indicated that DHB suppressed the degranulation of the IgE/BSA-stimulated BMCMCs by reducing the β-hexosaminidase release.

### 2.2. DHB Inhibited the Expression of FcεRI and the Binding of IgE to FcεRI

To identify whether DHB inhibits the expression of FcεRI and the FcεRI-IgE binding on the surface of BMCMCs, we performed flow cytometric analysis. DHB dose-dependently reduced the expression of FcεRI compared to that in the untreated control cells (Figure 2A). Typically, the binding of IgE on the surface of FcεRI initiates the activation of the mast cells and finally induces allergic reactions [5]. Based on the flow cytometry results (Figure 2B), IgE binding showed a marked increase in the IgE/BSA-stimulated cells compared with that of the control. Pretreatment with DHB significantly and dose-dependently reduced FcεRI expression at the investigated concentrations. DHB energy to FcεRI was −182.5245 kcal/mol. The binding site at the α-chain domain of DHB comprises amino acid residues, ARG D:427, ARG D:431, and LEU D:429 (Figure 2C–E). In addition, we confirmed that the binding energy of DHB to FcεRI-IgE was −206.748 kcal/mol, which means DHB was stably binding to FcεRI-IgE. In the given arrangement, the 4-hydroxy groups of DHB formed a covalent linkage with ARG D:431, which play a role in inhibiting the active site of FcεRI. Previous mutagenesis studies identified ARG D:427 as an important residue involved in IgE-FcεRI binding [22]. In contrast, the DHB binding site at the Cε3 domain comprises amino acid residues: ARG C: 334, CYS C: 335, VAL C: 336, ASP C: 362, LEU C: 363, ALA C: 364, LYS C: 367, HIS C: 422, HIS C: 424, LEU C: 425, and PRO C: 426. In the given arrangement, the 4-hydroxy group of DHB formed a covalent linkage and hydrogen bond interaction, respectively, with ARG C:334 and ALA C:364 of the Cε3 domain, which may assist in IgE inhibition. According to Garmen et al. (2000) [22], IgE binds to the FcεRI receptor at surface loops in Cε3, including the BC loop (ASP D:362–PRO D:365), DE loop (ARG D:393–THR D:396), FG loop (HIS D:424–ARG D:427), and the Cε2–Cε3 linker region (ASN D:332–VAL D:336) (Figure 2F–H). The present molecular docking results indicate that small molecular size and hydroxyl groups in DHB assist in the formation of a tighter bind with the active site pockets of FcεRI and IgE, thus inhibiting their binding. This leads to the suppression of IgE-mediated BMCMC degranulation. These results suggested that DHB led to the reduction of mast cell degranulation and the secretion of allergic mediators by downregulating the expression of FcεRI and the binding of IgE to FcεRI.

### 2.3. DHB Reduced the Secretion of Allergic Cytokines in the IgE/BSA-Stimulated BMCMCs

To further elucidate the effect of DHB on the secretion of allergic cytokines in IgE/BSA-stimulated BMCMCs. DHB treatment dose-dependently decreased the secretion of allergic cytokines such as IL-4, IL-5, IL-6, IL-13, and TNF-α at concentrations of 31.3 and 62.5 μg/mL (Figure 3). Interestingly, these results corresponded to the patterns of their gene expression levels (Appendix A). Collectively, we identified that DHB inhibited the allergic response, including mast cell degranulation and the secretion of allergic cytokines in IgE/BSA-stimulated BMCMCs.

### 2.4. DHB Inhibited the Activation of NF-κB Signaling Pathway in the IgE/BSA-Stimulated BMCMCs

The suppression of various cytokines expression via NF-κB inhibition is a key signaling pathway in the antiallergic effect [8]. Therefore, we investigated the effect of DHB on the IgE/BSA-induced action of NF-κB in the BMCMCs. As indicated in Figure 4, the stimulation of IgE/BSA induced the phosphorylation and degranulation of IκBα in the cytosol as well as the translocation of cytosolic NF-κB p65 to the nucleus. However, they were markedly modulated by the pretreatment of DHB in IgE/BSA-stimulated BMCMCs. Also, I.M, a positive control considerably reduced the activation of NF-κB p65. These results suggest that DHB improved the allergic response via inhibition of NF-κB signaling pathway in the IgE/BSA-stimulated BMCMCs.

### 2.5. DHB Improves the IgE/BSA Stimulation-Induced the PCA Reaction in Mice

To confirm the effect of DHB on the IgE/BSA stimulation-induced PCA reaction in mice, we monitored the amount of Evans blue dye extracted from the mice ear tissues. As shown in Figure 5A,B, stimulation by IgE/BSA markedly increased the PCA reaction by increasing the amount of Evans blue dye in mice ear tissue, compared to the normal mice group. Interestingly, it was significantly inhibited by the application of DHB with the reduced Evans blue amount. Next, we examined the degranulation of mast cells in mice ear tissues by histological analysis. The topical application of DHB dose-dependently suppressed the epidermal thickness and degranulation of mast cells in ear tissues (Figure 5C,D). This result indicates that DHB effectively improved allergic reactions.

## 3. Discussion

DHB, known to be a protocatechuic aldehyde and a water-soluble compound, is a phenolic aldehyde. Generally, phenolic compounds represent the main type of secondary metabolite in plants and show multiple biological effects [23]. Therefore, these are used as pharmaceuticals and/or functional food components as antioxidants, anti-inflammatory agents, and antiproliferation and antiallergic agents [24,25,26]. Several studies have reported the biological activities of DHB, particularly antioxidant activity, anti-cardiac hypertrophy, antiallergic effect, anti-inflammation, anti-atherosclerotic lesions, and anti-hepatitis B virus [17,19,24,25,26]. According to Kono et al. (2018), it was reported that DHB isolated from *Prunus mume* seed has antiallergy effect by suppressing the β-hexosaminidase release in antigen-stimulated RBL-2H3 cells and BMCMCs [20]. However, the specific biological mechanism underlying the effects of DHB has not yet been studied [27]. In the present study, we analyzed the antiallergic effect of DHB isolated from *P. morrowii*, a red alga, in IgE/BSA-stimulated BMCMCs and a PCA mouse model and evaluated its biological mechanism.

The annual increase in allergic diseases is caused by the inappropriate response of immune cells to allergens characterized by the excessive production of IgE [2,28]. IgE-allergenic stimulation is predisposed to produce histamine from stimulated mast cells through a degranulation process. With the releases of histamine, β-hexosaminidase, a granule-associated exoglycosidase, is simultaneously degranulated in the secretory granules of the mast cells and has been used as a biomarker of mast cell degranulation [5,28]. Here, we observed DHB reduced the release of β-hexosaminidase in IgE/BSA-stimulated BMCMCs. Treatment with 62.5 μg/mL DHB markedly suppressed the release of β-hexosaminidase compared with that of the only IgE/BSA-stimulated BMCMCs, without cytotoxicity. Therefore, we suggest that DHB suppresses mast cell degranulation by reducing β-hexosaminidase release. In a previous study, it was reported that DHB derived from *P. mume* seed inhibited β-hexosaminidase release upon antigen-induced degranulation of RBL-2H3 cells at the half maximal inhibitory concentration (IC_50_) values 0.6 mM, and it was similar to the effects of DHB on the degranulation of BMCMCs [20]. 

Generally, mast cells express FcεRI on their membranes, which are important for the proinflammatory allergic response, and when an IgE-antigen binds with FcεRI, the receptor is activated and complex biological reactions occur, causing allergic reactions, including inflammatory disorders [28]. Inhibiting IgE and FcεRI binding and FcεRI expression are some of the possible approaches to prevent effector cell activation [9]. Drugs that could achieve several or one of the above functions may be potential candidates for developing therapeutics [27]. Recently, pharmacological research has been interested in the isolation of natural active compounds showing antiallergic activity with demonstrated inhibition of β-hexosaminidase release as well as the expression of FcεRI and IgE binding to FcεRI, in marine algae [27,28]. Therefore, we chose the FcεRI and IgE as major target molecules and confirmed the inhibition activity of DHB on mast cell degranulation through flow cytometry and molecular docking analysis. Flow cytometric analysis is one of the best available methods to quantify the expression levels of specific surface receptor in live cells [9,10]. Molecular docking analysis is used in many natural product studies, because it can confirm the stable binding ability for interaction of natural compounds with various molecular targets [9]. Our results revealed that DHB dose-dependently reduced the levels of FcεRI expression on the surface of BMCMCs as well as IgE binding to FcεRI, compared to the IgE-stimulated BMCMCs. The IgE antibody Fc comprises domains (Cε2-Cε3-Cε4), whereas IgE readily binds with the FcCε3 domains of the receptor (the α-chain domain of FcεRI) [29]. Our previous study indicated BDB, a marine algal bromophenol of P. morrowii exerted anti-allergic effect as reducing the formation of IgE and FcεRI complex by binding to the active site of both FcεRI and IgE [9]. The outcomes of molecular docking analysis revealed that DHB led to the feasible values of −182.525 kcal/mol and −206.748 kcal/mol as free binding energies, provided that DHB could be a potent inhibitor of both FcεRI and IgE. DHB readily bound with the Cε3 domain of IgE than with the α-chain domain of FcεRI. These results clarify that the strong binding of the DHB to both FcεRI and IgE might reduce mast cell degranulation caused by the IgE/BSA stimulation.

Furthermore, the activation of mast cells leads to a potential source of many cytokines and a chemokine that might affect IgE-mediated allergic inflammation. These cytokines and chemokines include IL-1α, IL-2, IL-3, IL-4, IL-5, IL-6, IL-13, GM-CSF, TGF-β, TNF-α, IFN-γ, TARC, and several others [5,30]. IL-4 and IL-13 play an important role in mediating allergic reactions, which can promote the induction of IgE syntheses and the development of mast cells [5]. Particularly, IL-4 is the initial cytokine secreted from mast cells after IgE/BSA stimulation and activates the secondary cytokines, such as IL-5, IL-6, and IL-13 [31,32]. IL-5 is also known to have a specific action in the development, priming, and survival of eosinophils [33]. IL-6 had been defined as a modulate the adaptive immune response during early T-cell activation and TNF-α, which is a major effector cytokine in allergic reaction and improves mediator expression and cytokine in mast cells [2,34]. So, inhibition of these allergic cytokines and chemokines is one of the major indicators of relieved allergic responses. Thus, we confirmed that the secretion of allergic cytokines such as IL-4, IL-5, IL-6, IL-13, and TNF-α were down-regulated by the pretreatment of DHB in IgE/BSA-stimulated BMCMCs. With these results, we predict that DHB exerts an antiallergic effect by downregulating the gene expression and/or the protein secretion of various allergic cytokines and a chemokine as well as degranulation in IgE/BSA-stimulated BMCMCs.

The expression of cytokines is regulated by several intracellular signaling pathways, especially NF-κB, a transcription factor that plays a central role in the induction of gene expression and is considered as a promising target for the treatment of disease [5,35]. NF-κB represents a family of a prime transcription factors which regulate a variety of genes including different processes of the immune and inflammatory responses [36]. Activated NF-κB signaling results in the nuclear localization of NF-κB p65 and production of inflammatory cytokines (IL-1β, IL-6, and TNF-α) is increased. Especially, TNF-α activates the NF-κB signaling to amplify the response of inflammation [8]. Therefore, the inhibition of NF-κB is a major mechanism in mediating antiallergic effect. DHB significantly reduced the phosphorylation of IκBα and the translocation of cytoplasmic NF-κB p65 to the nucleus in IgE/BSA-stimulated BMCMCs. These results prove that DHB suppressed mast cell activation, cytokines production, and binding of the IgE to surface FcεRI via NF-κB signaling pathway.

Finally, we examined the anti allergic effects of DHB in a localized IgE-mediated allergic response based in vivo study. PCA reaction, which evaluate by increased vascular permeability and ear swelling, is that responses through binding of the IgE and FcεRI in mast cells [10,20]. Therefore, this model is generally used as the allergic animal model. Treatment of DHB inhibited the intensity of Evans blue dye in the ears. Thus, our results suggest that DHB exerts antiallergic effects through suppression of mast cell activation.

## 4. Materials and Methods

### 4.1. Materials

Minimum Essential Medium α (α-MEM), penicillin-streptomycin, fetal bovine serum (FBS), and dimethylsulfoxide (DMSO) were purchased from Thermo Fisher Scientific (Waltham, MA, USA). Pokeweed mitogen (PWM), 2-mercaptoethanol (2-ME), 3-(4,5-dimethylthiazol-2-yl)-2,5-diphenyltetrazolium bromide (MTT), Evans blue reagent, monoclonal anti-dinitrophenyl (DNP)-IgE (clone SPE-7), and β-actin antibody were purchased from Sigma-Aldrich (St. Louis, MO, USA). DNP-BSA was purchased from LSL Japan Inc. (Tokyo, Japan). Primary and secondary antibodies were purchased from Cell Signaling Technology Inc. (Danvers, MA, USA).

### 4.2. Purification of DHB from P. morrowii

DHB was purified from *P. morrowii* harvested from the coast of Jeju Island, according to the method described by Ko et al. (2019) [13]. Briefly, an 80% aqueous methanol crude extract of *P. morrowii* dry powder was suspended in water and fractionated into chloroform. The chloroform fraction was further separated on a silica open column via stepwise elution of chloroform and methanol mixtures of different ratios. The fractions with the highest bioactivity were pooled and further separated on a Sephadex LH-20 column with 100% methanol elution followed by reversed-phase (C18) HPLC purification (Alliance 2690; Waters Corp., Milford, MA, USA). The identity of the purified compound was confirmed using NMR. The purified DHB was diluted using PBS (2000-fold) for biological experiments (Appendix A).

### 4.3. Preparation of BMCMCs

C57BL/6 male mice (8-week-old), reared under specific pathogen-free conditions, were purchased from Orient Bio (Gwangju, Korea). BMCMCs were obtained from 8-week-old male C57BL/6 mice according to a previous method [10]. The cells were sub-cultured in α-MEM media (10% FBS, 1% penicillin-streptomycin) with 10% PWM-stimulated spleen cell-conditioned medium (PWM-SCM) and 0.2% 2-ME once a week for 7 weeks; and above 98% of the mast cells recognized as Giemsa-staining positive were used for the experiments. The institutional ethical committee has approved these experiments (Approval No. CNU IACUC-YS-2019-4) and was complying with “Principles of Laboratory Animal Care” guidelines (NIH publication No. 80-23, revised 1996).

### 4.4. Measurement of Cytotoxicity

The effect of DHB treatment on cell viability was evaluated using an MTT assay [9,10]. Accordingly, BMCMCs (2 × 10^4^/well) were treated with different concentrations of DHB (31.3 and 62.5 µg/mL) and incubated at 37 °C for 24 h. Subsequently, 15 μL of 5 mg/mL MTT solution was added, and the cells were further incubated for 4 h. The formazan crystals formed within the cells were dissolved using DMSO, and the absorbance was measured at 540 nm using a microplate reader (SpectraMax^®^ M2/M2e, Sunnyvale, CA, USA).

### 4.5. Measurement of β-Hexosaminidase Release

To assess the inhibitory effect of DHB on mast cell degranulation, BMCMCs (2 × 10^5^/well) were first incubated for 1 h with different DHB concentrations and sensitized using monoclonal anti-dinitrophenyl (DNP)-IgE (clone SPE-7) (1 μg/mL) for 4 h [25,26]. BSA-Tyrode’s buffer (0.1%) was used as a vehicle control where appropriate. Next, the BMCMCs sensitized with IgE were incubated for 1 h with the synthetic allergen, DNP-BSA, at 37 °C. After incubation, each supernatant was obtained, and the remaining cells were washed with 500 μL of 0.1% BSA-Tyrode’s buffer. The cells were then lysed in 50 μL of 0.5% Triton X-100-Tyrode’s buffer for 20 min. The respective supernatants and cell lysates were mixed with the substrate buffer (1.3 mg/mL 4-p-nitrophenyl-N-acetyl-β-D-glucosaminide and 0.1 M sodium citrate, pH 4.5) for 1 h. The reaction was terminated by adding 0.2 M glycine (pH 10.7), and the absorbance of the mixtures was measured at 405 nm using a microplate reader. β-hexosaminidase release from BMCMCs was calculated according to previously reported methods [26]. 

### 4.6. RT-PCR Analysis

Reverse transcription polymerase chain reaction (RT-PCR) was carried out using total RNA isolated from BMCMCs using Trizol reagent [37]. The cDNA was synthesized using PrimeScript RT Reagent Kit (TaKaRa Bio Inc., Shiga, Japan). PCR was carried out for 35 cycles using the respective primers indicated in our previous study [9]. The PCR conditions were set as follows: denaturation at 94 °C for 5 min, annealing at 55–60 °C for 1 min, and an extension phase at 72 °C for 20 min in a TaKaRa PCR Thermal Cycler (TaKaRa Bio Inc., Otsu, Japan). The PCR amplification products were electrophoresed on 1.5% ethidium bromide/agarose gels and visualized under UV transillumination (Vilber Lourmat, Marne la Uallee, France).

### 4.7. Measurement of Cytokines Levels

The BMCMCs (2 × 10^6^/well) were incubated for 1 h with different DHB concentrations and then sensitized with anti-DNP-IgE (1 μg/mL) for 4 h. Next, the sensitized BMCMCs were incubated in DNP-BSA at 37 °C for 24 h with a CO_2_ humidified atmosphere. The cell suspension supernatants were analyzed using mouse cytokine ELISA kits (Biolegend Inc., San Diego, CA, USA) according to the manufacturer’s instructions.

### 4.8. Flow Cytometric Analysis of Mast Cells Expressing FcεRI and FcεRI-IgE (IgE Binding)

To prepare the cells for the measurement of the surface expression levels of FcεRI, the BMCMCs (4 × 10^5^/well) were incubated with DHB for 6 h at 37 °C. To measure the surface expression levels of IgE- FcεRI binding, the cells (4 × 10^5^/well) were reacted with DHB for 2 h and sensitized using DNP-IgE (1 μg/mL) for 4 h at 37 °C. The prepared cells were blocked using anti-CD16/CD32 monoclonal antibodies (Thermo Fisher Scientific, Rockford, USA) for 20 min to prevent nonspecific binding of the antibodies. To detect the expression of FcεRI, the cells were incubated with FITC-conjugated anti-FcεRI antibodies (clone: MAR-1; Thermo Fisher Scientific, Rockford, IL, USA); to detect the expression of IgE-bound FcεRI, the cells were incubated with FITC-conjugated anti-mouse IgE monoclonal antibodies (clone: 23G3; Thermo Fisher Scientific, Rockford, IL, USA). The fluorescently stained cells were measured using flow cytometry (Beckman coulter, Brea, CA, USA). The total BMCMCs were gated on a forward scatter (FSC-A)/side scatter (SSC-A) plot to omit outliers.

### 4.9. Western Blot Analysis

The BMCMCs (2 × 10^5^/well) were incubated with DHB for 1 h before anti-DNP-IgE (1 μg/mL) sensitization, which was carried out for 4 h. Indomethacin (I.M) was used as a positive control. Then, the cells were treated with DNP-BSA (100 ng/mL) for 30 min. Cytosolic and nuclear proteins were separately obtained from the cells using a NE-PER^®^ nuclear and cytoplasmic extraction kit (Thermo Scientific, Waltham, MA, USA). The protein concentrations of the cell lysates were measured using a BCA protein assay kit (Thermo Scientific, Waltham, MA, USA) according to the manufacturer’s instructions. IκBα, phospho (p)- IκBα, p65, and p-p65 (1:1000 dilution, Cell Signaling Technology) and β-actin (1:3000 dilutions, Cell Signaling Technology) were used as the primary antibodies, and the secondary antibodies used were HRP-conjugated anti-mouse IgG and anti-rabbit IgG (1:3000 dilution, Cell Signaling Technology). The densitometry analysis of the Western blot results was performed using the ImageJ software to quantify the protein levels relative to corresponding housekeeping proteins [10]. 

### 4.10. In Silico Docking of DHB with IgE and FcεRI

The binding energies and types of intermolecular interactions between DHB with the active sites of IgE (PDB ID; 4GRG) and the IgE-FcεRI complex (PDB ID: 1F6A) were analyzed via molecular docking. The crystal structures of the respective proteins were obtained from the Protein Data Bank (PDB, http://www.pdb.org, accessed on 5 July 2017). The ligand (DHB) structure was optimized to a stable geometry via Gaussian semiempirical PM6 calculations using Gaussian 09 software. Ligand charge and torsional bonds were analyzed, and hydrogen atoms were added to both proteins and ligand before docking simulations. Flexible ligand docking and docking refinements were carried out using the CDOCKER plugin (default settings) in Accelrys Discovery Studio 3.5 (Accelrys, Inc., San Diego, CA, USA), which utilizes a molecular dynamics simulated-annealing-based algorithm. Potential binding sites for DHB were identified in the known binding pockets of the proteins.

### 4.11. PCA Test

Balb/C male mice (8-week-old), reared under specific pathogen-free conditions, were purchased from Orient Bio (Gwangju, Korea). The in vivo effects of DHB were analyzed in a PCA mouse model induced using DNP-IgE together with DNP-BSA. First, anti-DNP-IgE (500 ng) was intradermally injected into the dorsal skin of both ears. Then, using a 1 mL syringe, DHB was topical applicated to the mice 2 h before the induction of anaphylaxis. Subsequently, the mice were intravenously treated with 30 μL of DNP-BSA (10 mg) saline solution containing 4% Evans blue dye. The control group was treated with saline instead of DHB. After 30 min, the mice were euthanized by anesthetizing them with isoflurane followed by cervical dislocation, and skin tissues collected from the dorsal ear were soaked in 1 mL of formamide and left overnight (24 h) at 64 °C. The absorbance intensity at 620 nm was taken as a measure of the amount of extravagated dye. Histological analysis was conducted according to the method described previously [38]. The count of degranulated mast cells in ear tissues was obtained as described previously [39].

### 4.12. Statistical Analysis

The data were compared one-way ANOVA and Duncan’s multiple range test using the SPSS Statistics V20 software package (SPSS, Chicago, IL, USA). The values are presented as the means ± standard error (SE), and *p* values < 0.05 were considered significant.

## 5. Conclusions

In conclusion, the present findings demonstrated that DHB attenuated the antiallergic response through the inhibition of the mast cell degranulation, FcεRI expression, the crosslinking of IgE to FcεRI, and allergic cytokine secretion via the NF-κB activation in IgE/BSA-stimulated BMCMCs, and suppressed the allergic reactions in IgE/BSA-stimulation induced PCA model. DHB isolated from *P. morrowii* may be used as an agent for functional foods and pharmaceuticals to suppress allergic reactions.

## Figures and Tables

**Figure 1 marinedrugs-20-00133-f001:**
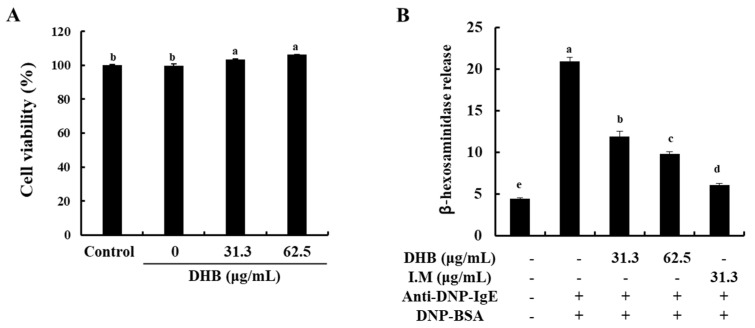
Effects of DHB on the cell viability and the β-hexosaminidase release of in IgE/BSA-stimulated BMCMCs. BMCMCs were incubated for 24 h with DHB (31.3 and 62.5 µg/mL) and I.M (indomethacin, a positive control). Effects of DHB on (**A**) the cell viability and (**B**) the mast cell degranulation were respectively measured using MTT and β-hexosaminidase release assay in BMCMCs. Values are expressed as means ± standard error (SE) of triplicate experiments. Bars with different letters (a–e) represent significantly difference (*p* < 0.05).

**Figure 2 marinedrugs-20-00133-f002:**
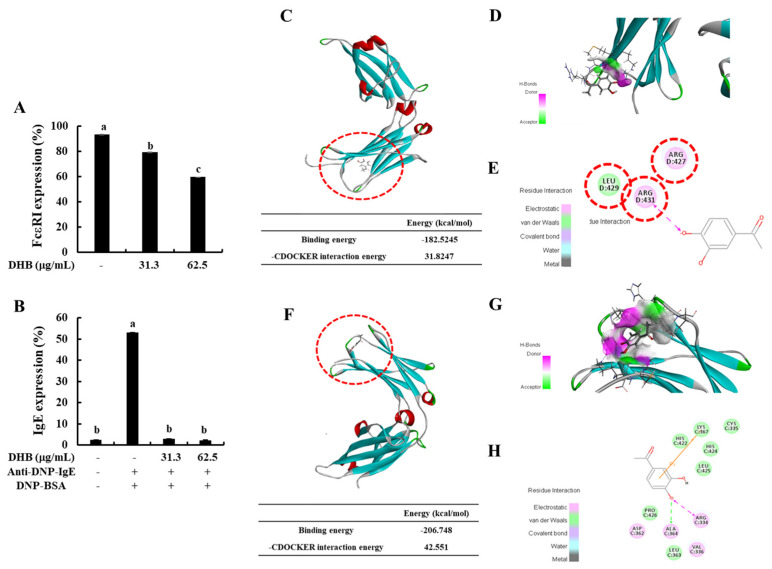
Effects of DHB on FcεRI expression and the binding of the IgE to FcεRI expression in IgE/BSA-stimulated BMCMCs and molecular docking analysis DHB binding. (**A**) Cell surface FcεRI expression and (**B**) IgE binding to FcεRI in BMCMCs were performed by flow cytometry. Prediction of stability and intermolecular interactions of DHB at (**C**–**E**) the active site of IgE and (**F**–**H**) IgE-FcεRI complex were identified by molecular docking analysis. Values are expressed as means ± standard error (SE) of triplicate experiments. Bars with different letters (a–c) represent significantly difference (*p* < 0.05).

**Figure 3 marinedrugs-20-00133-f003:**
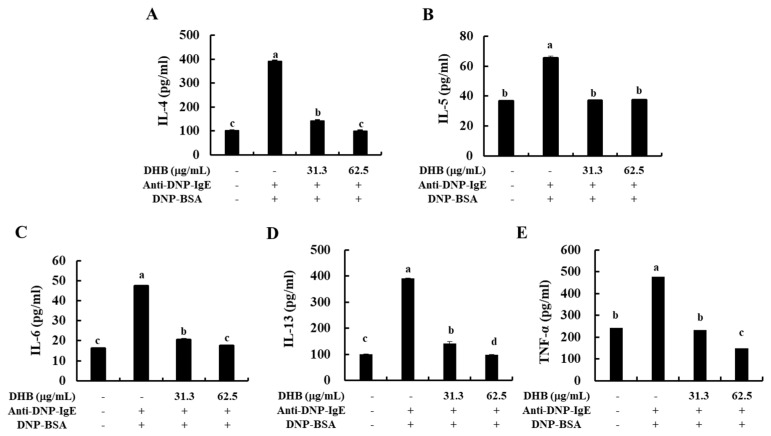
Effects of DHB on the secretion of allergic cytokines in IgE/BSA-stimulated BMCMCs. The secretion of (**A**) IL-4, (**B**) IL-5, (**C**) IL-6, (**D**) IL-13, (**E**) TNF-α in cell supernatants were measured by mouse cytokine ELISA kits. Values are expressed as means ± standard error (SE) of triplicate experiments. Bars with different letters (a–d) represent significantly difference (*p* < 0.05).

**Figure 4 marinedrugs-20-00133-f004:**
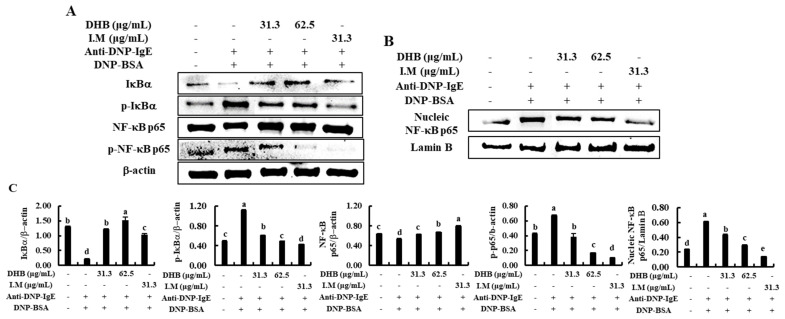
Effects of DHB on NF-κB signaling pathway in IgE/BSA-stimulated BMCMCs. (**A**) Cytosolic and (**B**) nucleic proteins obtained from BMCMCs were used for western blot analysis. The densitometry analysis of the western blot results was performed using (**C**) the ImageJ software to quantify the protein levels relative to corresponding housekeeping proteins. Values are expressed as means ± standard error (SE) of triplicate experiments. Bars with different letters (a–e) represent significantly difference (*p* < 0.05).

**Figure 5 marinedrugs-20-00133-f005:**
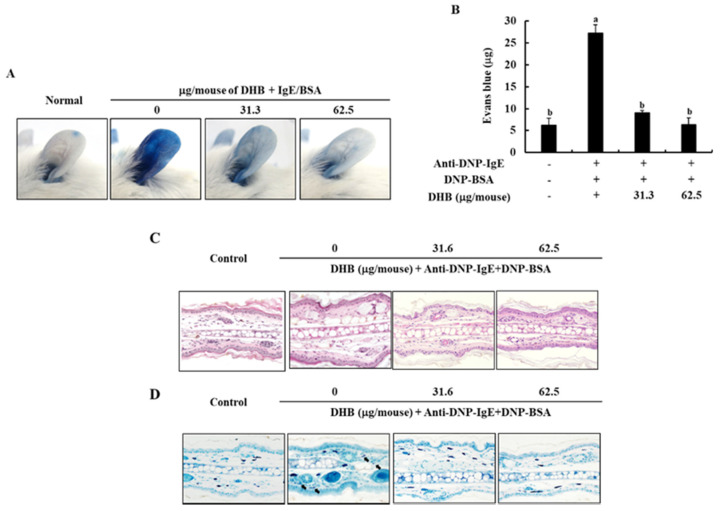
Effects of DHB on the IgE/BSA stimulation-induced PCA reaction in mice. (**A**) The photographic images of mice ear, (**B**) the amount of Evans blue dye extracted from mice ear tissues. (**C**,**D**) the degradation of mast cells was analyzed by H&E and toluidine blue staining assays in affected ear tissues. Values are expressed as means ± standard error (SE) of triplicate experiments. Bars with different letters (a, b) represent significantly difference (*p* < 0.05).

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
