# Peer review of "Anti-Allergic Effect of 3,4-Dihydroxybenzaldehyde Isolated from Polysiphonia morrowii in IgE/BSA-Stimulated Mast Cells and a Passive Cutaneous Anaphylaxis Mouse Model"

_marinedrugs, 2022, doi:10.3390/md20020133_

Round 1

Reviewer 1 Report

The manuscript reports the anti-allergic effect of DHB isolated from P. morrowii, a red alga, in IgE/BSA-stimulated BMCMCs and a PCA mouse model and evaluated its biological mechanism. It attenuated the anti-allergic response through the inhibition of the mast cell degranulation, FcεRI expression, the crosslinking of IgE to FcεRI, and allergic cytokine secretion via the NF-κB activation in IgE/BSA-stimulated BMCMCs, and suppressed the allergic reactions in IgE/BSA-stimulation induced PCA model. It could be accepted for publication in Marine Drugs after minor revisions.

  1. In the introduction, “Bis (3-bromo-4,5-dihydroxybenzyl) ether” was should be revised to “bis (3-bromo-4,5-dihydroxybenzyl) ether”.
  2. The 1H and 13 C NMR spectrum of DHB should be provided in the Supplementary data.
  3. For molecular docking analysis DHB binding. The binding of the small molecule ligand of the protein (IgE and IgE-FcεRI complex) should be provided, including its binding energy. Please explain why you chose the crystal structure protein of IgE (PDB ID; 4GRG) and the IgE-FcεRI complex (PDB ID: 1F6A).
  4. In the Figure 3, the a, b, c and d in the legend are easily confused with a, b, c and d in the figure. It is suggested to change them.

Author Response

We attached the answer about reviewer's comments as a file.

Thank you. 

Reviewer 2 Report

In this paper, the Authors presented new data on the biological mechanism underlying the anti-allergic effects of 3,4-dihydroxybenzaldehyde (DHB) in IgE-induced mast cell activation and in a PCA mice model.

They showed that the effect of DHB in attenuating allergic reactions is mediated by the inhibition of the mast cell degranulation, FcεRI expression, the crosslinking of IgE to FcεRI, and allergic cytokine secretion via the NF-κB activation.

This study gives new insight in the potential of DHB as therapeutic agent to mitigating allergic reactions, although it needs to be improved before publication.

Comments

  • General comments:

Author should revise the text carefully for a correct use of punctuation and for the style. Se, for example: Line 62; Line 73-74; Line 139.

  • Uniform the style for figure citation in the text.
  • Since DHB has been already tested in several model systems, as also mentioned in the discussion section, I suggest to improve the introduction to better describe the background knowledge about the bioactivity of this molecule.
  • Check the sentence at line 62, it seems that something is missing.
  • Paragraph 2.1 and 2.2 are too short in my opinion. Try to merge it indicating a new title for the merged paragraph.
  • In par. 2.3 the Authors report on the covalent linkage of DHB with the aminoacid residue ARG D:431 (line 110 and 116). I suggest to change this sentence in order to make it clearer that the linkage is only predicted according to docking analysis.
  • Figure 1. state the meaning of IM indicated in the figure.
  • Line 181-183: The sentence “This result indicates that DHB effectively suppressed allergic reactions in the IgE/BSA stimulation-induced PCA model” is repeated twice.
  • Figure S1 is not properly cited in the text. In addition, I suggest to add more information in the figure legend about how the results have been obtained.The NMR analysis must be added to Supplementary materials to support the identity and, more importantly, the purity of the purified compound used for the bioassays.
  • Please check the following assertion at the end of the Discussion: “In conclusion, the present findings demonstrated that DHB attenuated the anti-allergic response through the inhibition of the mast cell degranulation, FcεRI expression…”. Indeed, it seems in seems contrast to the claim that DHB could be a potential therapeutic candidate for the prevention and treatment of type I allergic disorders.
  • Revise the Reference list and format it in the correct style of the Marine Drugs Journal.

Author Response

(The authors gave the same response as above.)

Round 2

Reviewer 1 Report

The authors have revised the manuscript according to the previous comments. Now it can be accepted for publication.

Author Response

Thank you for your evaluation

Reviewer 2 Report

The Authors improved their manuscript according to the questions raised by the reviewers. There are no additional issues to be addressed.

Author Response

Thank you for your evaluation